# Catalytic Gasification of Petroleum Coke with Different Ratios of K_2_CO_3_ and Evolution of the Residual Coke Structure

**DOI:** 10.3390/molecules28196779

**Published:** 2023-09-23

**Authors:** Man Zhang, Hongyu Ban, Zhiqing Wang, Xinning Xiang, Xiaolei Wang, Qian Zhang

**Affiliations:** 1School of Chemical Safety, North China Institute of Science and Technology, Langfang 065201, China; zhang2597097@126.com; 2The System Design Institute of Mechanical-Electrical Engineering, Beijing 100854, China; zhaoxiaoyu_8422@163.com; 3State Key Laboratory of Coal Conversion, Institute of Coal Chemistry, Chinese Academy of Sciences, Taiyuan 030001, China; 4State Key Laboratory of Clean and Efficient Coal Utilization, Taiyuan University of Technology, Taiyuan 030024, China; xiangxinning1017@163.com (X.X.); w19581599650@163.com (X.W.); zhangqian01@tyut.edu.cn (Q.Z.)

**Keywords:** petroleum coke, catalytic gasification, K_2_CO_3_, char structure, evolution

## Abstract

The catalytic gasification of petroleum coke with different ratios of K_2_CO_3_ was investigated by a thermogravimetric analyzer (TGA) using the non-isothermal method. The initial, peak, and final gasification temperatures of the petroleum coke decreased greatly as the amount of K_2_CO_3_ increased, and the catalytic reaction became saturated at a concentration of K^+^ higher than 5 mmol/g; with the further increase in catalyst; the gasification rate varied slightly, but no inhibition effect was observed. The vaporization of the catalyst was confirmed during the gasification at high temperatures. The structural evolution of the residual coke with different carbon conversions was examined by X-ray diffraction (XRD), Raman spectroscopy, and N_2_ adsorption analyses during gasification with and without the catalyst. The results showed that the carbon crystallite structure of the residual coke varied in the presence of the catalyst. As the carbon conversion increased, the structure of the residual coke without the catalyst became more ordered, and the number of aromatic rings decreased, while the graphitization degree of the residual coke in the presence of the catalyst decreased. Meanwhile, the surface area and pore volume of petroleum coke increased in the gasification process of the residual coke, irrespective of the presence of the catalyst. However, the reactivity of the residual coke did not change much with the variation in the carbon and pore structure during the reaction.

## 1. Introduction

As the upgrade of heavy petroleum has expanded, together with a gradual growth in oil demand, the amount of petroleum coke produced in the petroleum refining process has increased rapidly [1,2]. Due to its high content of sulfur and transition metals such as V and Ni, petroleum coke is not a good fuel for direct combustion [3]. Compared with combustion, the gasification of petroleum coke can produce syngas that could be used for the synthesis of chemical products or H_2_ [4]. However, the high graphitization degree and undeveloped pore structure caused a relatively low gasification reactivity of the petroleum coke [1,3]. Increasing the reaction temperature could promote the gasification of petroleum coke, but these severe reaction conditions increase the energy consumption and the cost of the process. As reported, the addition of alkali or alkaline earth metallic species, such as K_2_CO_3_, Na_2_CO_3_, KOH, NaOH, and Ca(OH)_2_, could greatly improve the gasification reactivity and reduce the gasification reaction temperature by decreasing the oxygen demand and greatly enhancing the thermodynamic efficiency [5]. Accordingly, potassium carbonate (K_2_CO_3_) is the most preferable catalyst, which can significantly improve the gasification reaction at low temperatures [5,6]. In recent years, the catalytic gasification of coal, biomass, or other carbonaceous materials has been widely investigated [1,5,6]. However, it was found that during gasification in the presence of a catalyst, the alumina and silica species contained in coal or biomass ash will react with the alkali or alkaline earth metallic species, forming inactive aluminosilicate minerals that cause the deactivation of the catalyst [7]. Unlike coal or biomass, petroleum coke contains quite limited amounts of Si or Al minerals; thus, catalytic gasification could be an optimal method for its utilization.

The catalytic effect of K_2_CO_3_ during the catalytic gasification of petroleum coke depends on many factors, including the amount of catalyst that is added and the reaction temperature [5,8]. Though a high temperature would promote the endothermic gasification reaction and increase the mobility of K_2_CO_3_, leading to its good dispersion in the sample and increasing the catalytic effect, the primary purpose of catalytic gasification is to improve the gasification efficiency while lowering the operating temperature. Moreover, potassium might vaporize at very high temperatures, which would reduce the catalytic effect [7,9]. Thus, increasing the catalyst amount is believed to be a better way to improve the gasification reactivity as it would provide more active intermediates on the carbon surface. Some researchers reported that during catalytic gasification, the catalyst saturation level could be reached with the additional amount increase, above which a decrease in the gasification rate is observed [7]. An excess of catalyst blocks the pores in the carbon sample, restricting access to the gasifying agent [6,8], so it is important to determine the catalyst saturation level. In addition, the mixing procedure used to combine the catalyst and the petroleum coke is also important, as it affects the dispersion of the catalyst in the coke and their degree of contact. Ion exchange and impregnation are considered better mixing methods than physical mixing [10], but physical mixing is more cost-effective and is easier to implement in large-scale experiments. In addition, some researchers reported that achieving a homogeneous catalyst dispersion is not important when using potassium (K), as this catalyst is mobile at high temperatures in the presence of carbon [11]. Therefore, physical mixing was used in this study.

Thermogravimetric analysis (TGA) is widely used for studying the gasification behavior of materials as it is very sensitive and can be performed in well-controlled experimental conditions. It can be conducted in isothermal and non-isothermal modes [12]. In the isothermal mode, sample gasification is performed at a given temperature, testing several temperatures to precisely define the kinetic characteristics of the process. In contrast, non-isothermal gasification is conducted by varying the temperature during the process according to a preset program. The non-isothermal method is simpler than the isothermal one; it would not cause huge changes in the chemical and physical properties of the tested sample during the heating stage and could provide more information through fewer experiments [12]. In particular, during catalytic gasification, a non-isothermal analysis could better evaluate the catalytic behavior of the examined sample by a series of parameters such as the initial, peak, and final temperatures. For these reasons, non-isothermal gasification was employed in this study. As most reported gasification experiments were performed using the isothermal procedure, it is still uncertain whether the catalytic saturation or the inhibition effect would occur in non-isothermal conditions. Meanwhile, in the non-isothermal mode, changes in the carbon structure of the residual coke can occur for several reasons, such as a partial consumption of coke, an increase in the reaction temperature, and the interactions between the carbon and the catalyst; all these phenomena will affect the reactivity of the residual coke and, finally, the efficiency of the process; thus, the variation in the residual coke also needs to be considered.

This study aimed to evaluate the catalytic gasification behavior of petroleum coke with a wide range of K_2_CO_3_-adding concentrations. The catalytic gasification performance under different catalyst concentrations, the catalyst saturation level, and the vaporization of the catalyst were examined and discussed. Then, the detailed characteristics and reactivity of residual coke samples with different carbon conversions obtained from gasification with or without catalysts were analyzed and discussed.

## 2. Results and Discussion

### 2.1. Catalytic Gasification Reactivity of Petroleum Coke

Figure 1 shows the mass loss and the mass loss rate curves of the CO_2_ gasification of ZH petroleum coke with different K_2_CO_3_ amounts. In Figure 1, the mass loss observed at low temperatures (less than 700 °C) was mainly caused by the release of residual volatiles contained in ZH; as the temperature increased, CO_2_ gasification started, and the rate of mass loss increased quickly. For most of the samples, a single peak is present in the mass loss rate curves, after which the mass loss rate decreases until the end of the reaction. The characteristic parameters of the initial, peak, and final reaction temperatures are listed in Table 1. The addition of K_2_CO_3_ obviously decreased the characteristic temperatures of the gasification reaction. With the increase in K_2_CO_3_ loading (corresponding to an amount of K^+^ lower than 3 mmol), the mass loss curve shifted to a lower-temperature region, and the peak temperature required to reach the maximum reaction rate also decreased significantly. At the temperature of 1000 °C, the mass loss of ZH without the catalyst was less than 20%, while in the presence of a catalyst amount higher than 1 mmol K^+^, it increased to 99%, indicating that most of the carbon had reacted with CO_2_ and that the addition of K_2_CO_3_ greatly improved the gasification reactivity of petroleum coke. With the temperature increase, the potassium rapidly diffused to the coke surface, promoting the formation of pores or channels that increased the exposure of the carbon atoms [13,14]. On the other hand, K_2_CO_3_ reacted with coke, forming -COK or -CK surface intermediates [6,15], which could easily acquire oxygen from CO_2_ and transfer it to the surface of the carbon to form CO. With the catalyst concentration increase, the amounts of active intermediates and pores or channels on the carbon surface increased, thus enhancing the gasification reactivity.

As the catalyst amount further increased (above 3 mmol of K^+^), though the initial and peak temperature still decreased gradually, their reduction was obviously more limited, as shown in Figure 1 and Table 1. Moreover, when the K^+^ amount was higher than 5 mmol, the reduction in the initial temperature was quite small, indicating that the catalytic effect had come to a saturation [5,11]. This might be attributed to the excessive catalyst deposition on the surface of petroleum coke, which prevented the formation of active intermediates and the increase in the gasification reaction. However, in contrast to the catalytic gasification of coal or biomass reported, no inhibition effect was observed on the catalytic gasification of ZH. Even for the sample with 14 mmol K, in which the mass percentage of K_2_CO_3_ in the coke was 49.14%, the gasification curve was still very similar to that of the 5 mmol K sample. The main reason might be that in non-isothermal mode, the initial temperature was much lower than the melting point of K_2_CO_3_ (891 °C); thus, the carbon pores on the coke surface were not blocked, allowing the gasifying agent to enter the pore structure of petroleum coke and preventing an increase in the diffusion resistance. The second reason might be that the quantity of minerals contained in petroleum coke was much lower than that present in coal or biomass. In fact, limited levels of Si or Al present avoided the catalyst deactivation by the formation of inactive aluminosilicate minerals during the reaction [7]. Moreover, the results obtained at the initial temperature of 700 °C, which is much lower than the melting point of K_2_CO_3_ (891 °C), suggested that melting was not essential for the catalytic effect to occur.

From the mass loss curves shown in Figure 1, it also appears that the final masses of the samples were much lower than the initial blending mass of K_2_CO_3_. This indicated that some of the catalysts vaporized or decomposed during the reaction at high temperatures. For the samples with a K^+^ quantity lower than 3 mmol, the mass loss and mass loss rate curve did not allow for the distinction between carbon gasification and catalyst vaporization. In contrast, for the samples with an amount of K^+^ higher than 5 mmol, besides the obvious mass loss, a second small peak was observed, which appeared at temperatures higher than 950 °C and was probably caused by the vaporization of potassium. As reported, at high temperatures above 750 °C, incongruent vaporization of K_2_CO_3_ could be observed, and the vaporization was affected by the presence of residual carbon and the vapor pressure of CO_2_ [10]. Usually, the vaporization of the catalyst is not considered convenient, as it is associated with a decreased catalytic effect and a loss of the catalyst [11]. From our point of view, if the vaporized potassium could be recovered by applying different temperature stages in the reactor [16], then the catalyst could be recycled.

Based on the above analyses, in the catalytic gasification in non-isothermal mode, the coke reacted with the gasifying agent as the temperature increased, and the catalyst increased the gasification rate by facilitating oxygen transfer cycles. The residual carbon structure of coke could be affected by the temperature, the gasification reaction, and the catalyst; how the carbon structure changes during the gasification and whether it will affect the reactivity of the residual coke is still unclear. Therefore, the detailed characteristics and reactivity of the residual coke with different carbon conversions were analyzed.

### 2.2. XRD Analysis of the Residual Coke

The XRD spectra of the petroleum coke without the catalyst and that with 5 mmol of K^+^ for different carbon conversion values are shown in Figure 2a,b, respectively. The preparation procedure of the residual coke is illustrated in Section 3.3, and the residual coke obtained was named according to its carbon conversion value. Two typical peaks, denoted as 002 band (diffraction angle at approximately 25°) and 100 band (at approximately 43°), appeared. The 002 band could be attributed to the content of crystalline carbon, while the 100 band could be attributed to the degree of condensation of aromatic rings in the samples. The amorphous and microcrystalline structures in the samples were the main features responsible for the production of background intensities and areas in the diffraction peaks [17,18]. As shown in Figure 2a, with the gradual increase in carbon conversion, the intensity of the 002 band of the residual coke increased significantly, and the diffraction peak narrowed. By contrast, as shown in Figure 2b, the 002 band of the residual coke obtained with 5 mmol of K^+^ became smaller and widened, indicating that the graphitization structure was gradually consumed during the catalytic gasification process.

To further analyze the structural features, the 002 band and the 100 band were further curve-fitted, and the crystallite size, including the stacking height (L_c_), the interplanar spacing (d_002_), the crystalline diameter (L_a_), and so on [14], was calculated, as shown in Table 2. The trends of the variation in L_a_ and L_c_ with different carbon conversions are shown in Figure 3. Compared with the residual coke without the catalyst, the value of L_c_ decreased from 13.72 Å to 9.70 Å (Table 2) during catalytic gasification, implying that more active carbon or small-aromatic-ring systems were generated. The potassium catalyst addition caused the crystalline structure of the residual coke to become more disordered and reduced the graphitization degree. In other words, as the reaction proceeded, the catalyst restrained the rearrangement of the graphene layers, enhancing the interfacial defects between adjacent basic structural units. The catalyst could be inserted into the edge of the aromatic rings, resulting in internal lattice defects [19,20]. However, for the residual coke without the catalyst, the L_a_ value increased, indicating that the growth of crystallites was promoted as the carbon conversion and the temperature increased [17]. Therefore, it could be concluded that in the non-isothermal gasification process, the amorphous carbon and microcrystalline structure of the residual coke samples underwent different changes depending on the addition of the catalyst. In addition, as the graphitization degree of carbon is always inversely proportional to coke reactivity, the catalyst addition might improve the reactivity of the residual coke by reducing the graphitization degree.

### 2.3. Raman Spectra of the Residual Coke

Raman spectrometry is highly sensitive for the detection of crystal structures and amorphous structures in carbon-containing materials. Figure 4 shows the Raman spectra of ZH with different carbon conversions in the region of 800–1800 cm^−1^. Two distinct peaks at 1580 and 1350 cm^−1^, named G band and D band, were further curve-fitted into five Gaussian bands [21], and the obtained parameters were compared in Figure 5. The D_1_ band (peak at 1350 cm^−1^) was attributed to atomic defects. The D_2_ band (1620 cm^−1^) corresponded to the surface of the graphite layer. The D_3_ band (1500 cm^−1^) originated from the amorphous carbon in the carbon structure. The D_4_ band (1200 cm^−1^) mainly indicated the volatile hydrocarbons and the disordered graphite lattice. The G band (1580 cm^−1^) represented in-plane vibrations of aromatic carbons in the graphitic structure. The value of the I_D1_/I_G_ ratio (ratio of the areas of the D1 and G bands) was inversely related to the planar dimensions of the crystal; a decrease in the I_D1_/I_G_ ratio indicated an increase in the ordering of the carbon structure, while an increase in the I_G_/I_All_ ratio indicated an increase in the order degree of the carbon structure [22].

As shown in Figure 5a, the I_D1_/I_G_ ratio for the residual coke without the catalyst decreased during gasification. This indicated that the concentrations of aromatic rings with six or more fused benzene rings increased during gasification as a result of the dehydrogenation of hydro-aromatics and the growth of aromatic rings [23]. However, for the residual coke with the catalyst, the I_D1_/I_G_ ratio increased with the carbon conversion. The catalyst could reduce the development of large aromatic rings during CO_2_ gasification. Therefore, the crystal structure of the residual coke became disordered, and the catalyst could restrain the graphitization process by weakening the delocalized π bonds in its interaction with the coke [19]. Meanwhile, during catalytic gasification, the values of the I_G_/I_All_ ratio decreased, which was consistent with the XRD analysis reported above, indicating that a continual disordering process occurred. It is worth noting that for the residual coke without the catalyst, the values of the I_G_/I_All_ ratio increased, which was attributed to the fact that the active carbon was consumed first, and a greater extension of the graphite-like structure was lost [18]. In addition, the activation effect of the gasifying agent was quite limited.

### 2.4. Pore Structure Analysis of the Residual Coke

The pore structure of the samples analyzed by N_2_ adsorption is shown in Figure 6, and the corresponding parameters are listed in Table 3. As illustrated in Figure 6, the N_2_ adsorption–desorption isothermal curves of residual coke with different carbon conversions showed the H_3_ or H_4_ type hysteresis loop, indicating that an irregular pore structure formed during the reaction. The pore size distribution curves also showed that the micro and meso pores were well-developed with the carbon conversion increase. For the residual coke obtained from the gasification of ZH without or with the catalyst under different conversions, the surface area and the pore volume increased, as shown in Figure 7. The surface area increased, respectively, from 1.81 m^2^/g to 99.90 m^2^/g and from 4.01 m^2^/g to 72.17 m^2^/g, and the pore volume increased, respectively, from 0.008 cm^3^/g to 0.121 cm^3^/g and from 0.013 cm^3^/g to 0.054 cm^3^/g, as reported in Table 4. It is reasonable to infer that closed pores opened, and new pores formed in the early stage of the gasification reaction, resulting in a significant increase in the surface area and pore volume [18,24]. The gasifying agent, gasification temperature, and the catalyst might also promote the formation of the pores during gasification, inducing a gradual increase in the surface area and pore volume [24,25]. During the non-isothermal gasification reactions, the residual coke obtained from ZH with the addition of catalyst was prepared at a lower temperature during a similar carbon conversion compared with that of residual coke without the addition of catalyst; thus, it did not show a significant difference in the pore structure variation.

### 2.5. Residual Coke Reactivity Analysis

Figure 8 shows the combustion behavior of the residual coke obtained from the gasification of ZH without or with the catalyst for different conversions; the characteristic temperatures are listed in Table 4. For the residual coke obtained from the gasification of ZH without the catalyst, shown in Figure 8a,b, the combustion reactivity of the samples for different conversions was quite similar; the more ordered carbon structure and increased surface area did not affect the reactivity. The raw ZH coke had the highest combustion reactivity, which might be attributed to the accelerating effect of the volatiles it contained [26]. For the residual coke obtained from the gasification of ZH with the catalyst shown in Figure 8c,d, as the carbon conversion increased, the initial temperature remained almost unvaried, but the peak temperature decreased, and a second peak at about 600 °C appeared for residual cokes with the conversion of 48% and 65%. It is hard to differ the reactivity of the residual coke increase or decrease as the carbon conversion increases. Based on the residual coke structure analyses, the graphitization degree decreased, and the surface area and pore volume increased, which would increase the reactivity, as reported [8,10,27]. Then, it could be concluded that the variation in the carbon and pore structure was not the key factor affecting the residual coke’s reactivity. Moreover, compared with the reactivity of residual cokes with different conversions without the catalyst, that of residual cokes in the presence of the catalyst was much higher, indicating that the reactivity of coke was significantly altered by the addition of the catalyst, and this could be due to the interactions between the petroleum coke and the catalyst, which needs to be studied further.

## 3. Materials and Methods

### 3.1. Samples

Petroleum coke, obtained from Sinopec Zhenhai Refining and Chemical Company and denoted as ZH, was used in this study. The sample was ground and sieved to a size below 0.074 mm before use. The proximate and ultimate analysis of ZH are shown in Table 5. K_2_CO_3_ with a purity >99.0% was obtained from Sinopharm Chemical Reagent Co., Ltd., China. K_2_CO_3_ was added to ZH by physical blending. The K^+^ concentration in petroleum coke was in the range of 0.1~14 mmol K^+^/g. For example, 1 g of ZH coke was mechanically blended with 2.5 mmol of K_2_CO_3_ (0.345 g) in a mortar for 10 min, and the prepared sample was designated as with 5 mmol K, reflecting that 5 mmol of K^+^ was added for 1 g of petroleum coke. Using this method, a series of samples, named With 0.1 mmol K, With 0.5 mmol K, With 1 mmol K, With 2 mmol K, With 3 mmol K, With 5 mmol K, With 10 mmol K, With 12 mmol K, and With 14 mmol K, were prepared, as reported in Table 6. All samples were evenly mixed and stored for use.

### 3.2. CO_2_ Gasification

The non-isothermal catalytic gasification reactivity measurements were conducted on a Setaram Setsys TGA analyzer. In each gasification experiment, about 2 mg of the sample was loaded evenly in an alumina crucible and heated at 10 °C/min under a CO_2_ atmosphere (100 mL/min) until no significant mass change in the sample was recorded. The prior reaction conditions had eliminated the effect of internal and external diffusion. In the final experimental data, the influence of the change in gas buoyancy during non-isothermal gasification was eliminated by subtracting a blank run.

In the non-isothermal catalytic gasification process, the carbon conversion X of the samples was calculated as follows [11,28]:(1)$$X=\frac{W_{0}-W_{t}}{W_{0}-W_{f}}$$
where W_0_ represents the initial mass; W_t_ represents the instantaneous mass at reaction time t, and W_f_ represents the final mass of the petroleum coke after complete conversion.

### 3.3. Residual Coke Preparation

Residual coke with different carbon conversions was prepared on a fixed-bed reactor. A schematic of the experimental setup is presented in Figure 9. The procedure for the residual coke preparation was as follows: about 1 g of ZH coke or the sample of ZH coke with 5 mmol K was weighed, evenly spread in an alumina crucible, and then placed into a furnace. CO_2_ was used to replace the air in the furnace tube for about 30 min. Then, the furnace was heated at a rate of 10 °C/min in the CO_2_ atmosphere until it reached the preset temperature. Subsequently, the crucible was transferred to the low-temperature zone to cool down the coke under the CO_2_ atmosphere. After that, the cooled sample was ground and stored in a dryer. To avoid the influence of the catalyst on the characterization of the carbon structure of the residual coke, the samples were washed with deionized water until a neutral pH was attained. After that, the residual coke was dried in a vacuum oven at 110 °C for 24 h and packaged in a desiccator for further use. By presetting the final temperature and weighing the residual coke mass, the carbon conversion could be calculated; the residual coke was named by its carbon conversions.

### 3.4. X-ray Diffraction Analysis

The structural characterization of the coke samples was performed by a Rigaku Ultima IV powder X-ray diffraction meter (XRD) using Cu Kα radiation (λ = 1.5418 Å) that operated at 40 kV and 30 mA over the range from 10° to 90° at a scanning speed of 5 °/min.

### 3.5. Raman Spectroscopy

A Raman spectrometer (Horiba HR800, HORIBA Scientific, Kyoto, Japan) was used to determine the carbon microcrystalline structural parameters of the residual coke with different carbon conversions. The aromatic structure of the samples was recorded in the range of 800–2000 cm^−1^.

### 3.6. Pore Structure

A surface analyzer (Quantachrome QDS-30, Quantachrome Instruments, Florida, USA) was employed to determine the evolution of the coke samples with different carbon conversions. N_2_ at 77 K was used as the adsorbed gas. The coke samples were outgassed at 200 °C in a vacuum for 5 h before each N_2_ adsorption–desorption experiment. The surface area and pore volume of the samples were calculated by the BET equation and the BJH method [24].

### 3.7. Residual Coke Activity Evaluation

The reactivity of the residual coke samples was detected by the Setaram Setsys TGA analyzer. The residual coke was washed in deionized water and dried. The procedure was as follows: about 2 mg of the sample was weighed, evenly spread in an alumina crucible, and then heated at 10 °C/min in the air (100 mL/min) until no obvious mass change was detected. The effect of K_2_CO_3_ on the gasification reactivity of ZH with different carbon conversions was verified by this method.

## 4. Conclusions

In this paper, the catalytic gasification of petroleum coke with K_2_CO_3_ was investigated, and the structural evolution of the residual coke with different carbon conversions was examined. The results showed that the initial, peak, and final gasification temperatures of the petroleum coke decreased greatly with an increase in the catalyst ratio, and the catalytic reaction was saturated when the K^+^ concentration was 5 mmol/g; at this and higher catalyst concentrations, the gasification rate varied slightly. K_2_CO_3_ had an excellent catalytic effect on the gasification of ZH and did not exert any inhibition effect even when the catalyst concentration was higher than the saturation concentration. The vaporization of the catalyst was observed during the reaction at high temperatures.

The carbon crystallite structure of the residual coke underwent different changes in the presence of the catalyst. As the carbon conversion increased, the structure of the residual coke without the catalyst became more ordered, and the number of aromatic structures with six or more fused benzene rings increased. In contrast, for the residual coke obtained from catalytic gasification, the graphitization degree decreased, and the crystal structure became disordered as the carbon conversion increased. The surface area and pore volume of petroleum coke increased in the gasification process of the residual coke with or without a catalyst. However, the reactivity of the residual coke did not change much with the variation in the carbon and pore structure during the reaction, indicating that the pore and carbon structure were not the key factors affecting the reactivity. The catalytic gasification of petroleum coke is a promising method for disposing of petroleum coke. It is suggested that further investigations of the interactions between petroleum coke and catalysts should be conducted in the future.

## Figures and Tables

**Figure 1 molecules-28-06779-f001:**
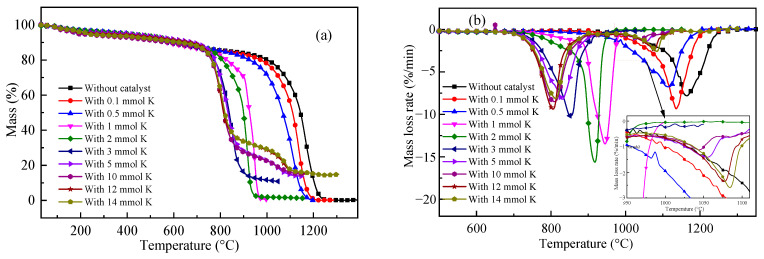
Gasification of petroleum coke in the presence of different amounts of K_2_CO_3_: (**a**) mass variation as a function of the temperature; (**b**) mass loss rate variation as a function of the temperature.

**Figure 2 molecules-28-06779-f002:**
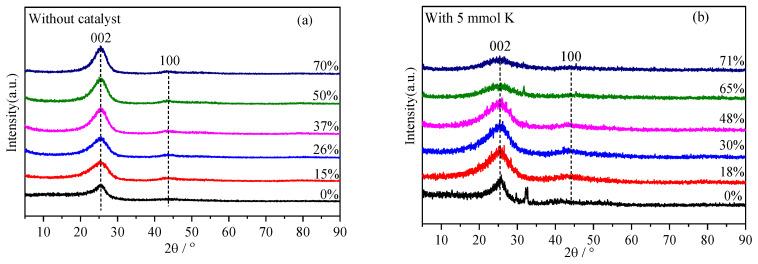
XRD spectra of residual coke with different carbon conversions.

**Figure 3 molecules-28-06779-f003:**
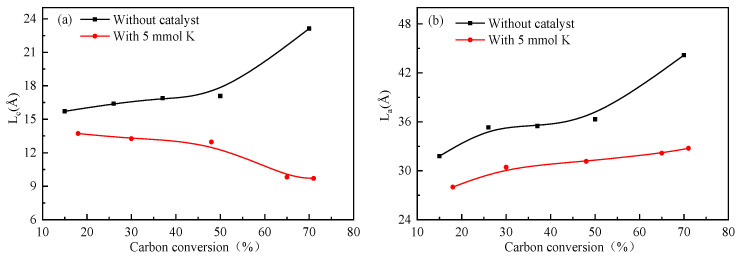
Comparison of the XRD parameters of residual coke with different carbon conversions. (**a**) Lc. (**b**) La.

**Figure 4 molecules-28-06779-f004:**
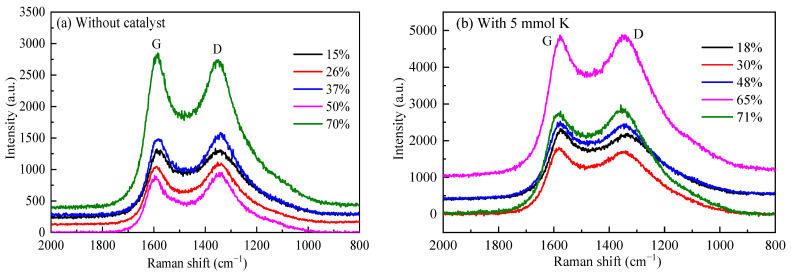
Raman spectra of residual coke with different carbon conversions.

**Figure 5 molecules-28-06779-f005:**
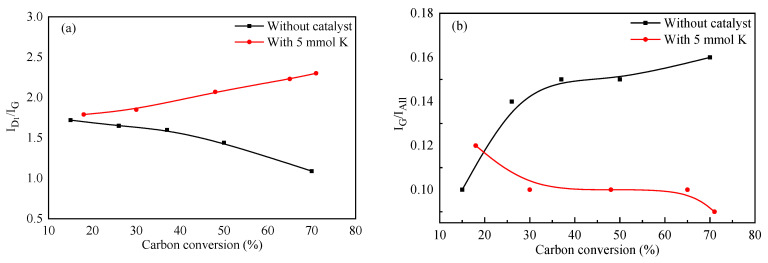
Comparison of the ratios of the areas of different bands of residual coke with different carbon conversions. (**a**) I_D1_/I_G_ and (**b**) I_G_/I_All_.

**Figure 6 molecules-28-06779-f006:**
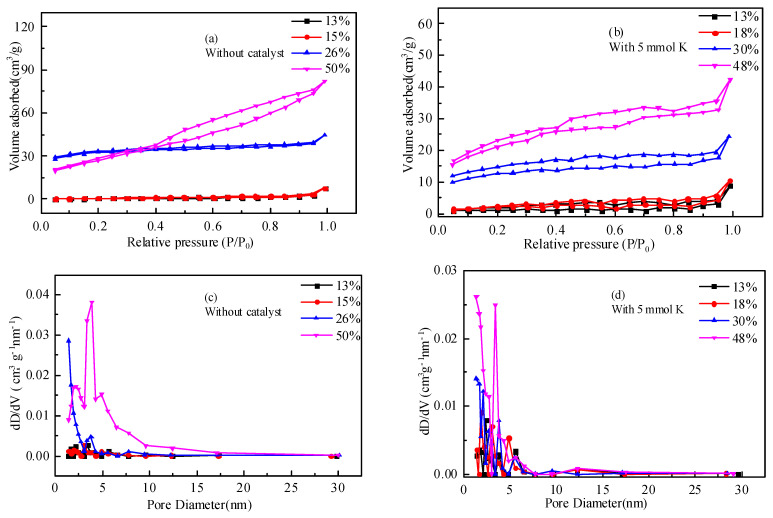
N_2_ adsorption–desorption isothermal curves and pore size distribution curves of residual coke with different carbon conversions.

**Figure 7 molecules-28-06779-f007:**
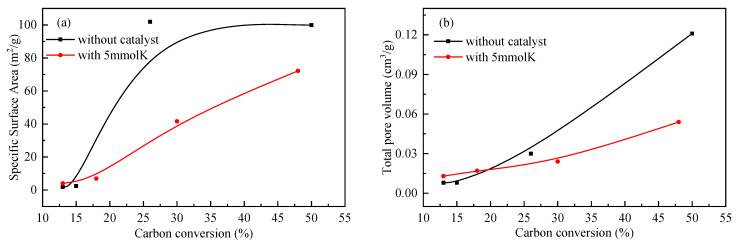
Comparison of the surface area and pore volume of residual coke with different carbon conversions. (**a**) Surface area. (**b**) Pore volume.

**Figure 8 molecules-28-06779-f008:**
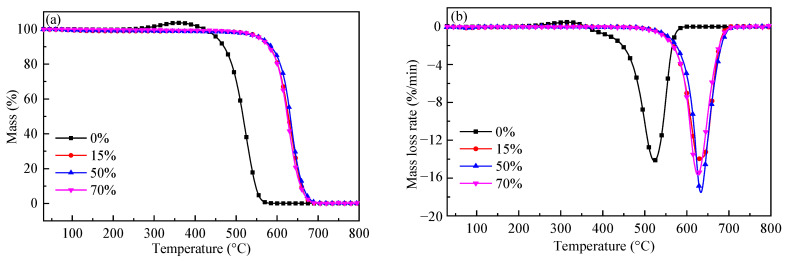
Combustion curves of residual coke with different carbon conversions. (**a**,**b**) Without catalyst. (**c**,**d**) With 5 mmol K.

**Figure 9 molecules-28-06779-f009:**
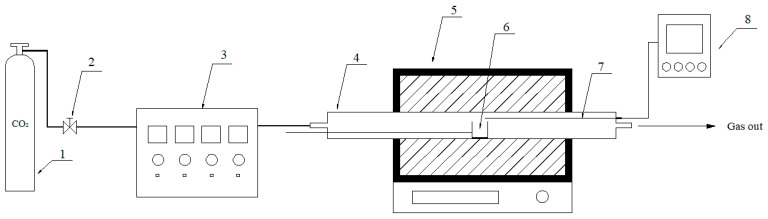
Schematic diagram of the fixed-bed reactor system. 1. High-pressure vessel. 2. Relief valve. 3. Gas flowmeter. 4. Stainless furnace tube. 5. Tube furnace. 6. Sample. 7. K-thermocouple. 8. Temperature indicator.

**Table 1 molecules-28-06779-t001:** The characteristic temperature of gasification of petroleum coke with different K_2_CO_3_ amounts.

Sample	Temperature (°C)		
	T_i_	T_m_	T_f_
Without catalyst	930	1163	1250
With 0.1 mmol K	940	1136	1213
With 0.5 mmol K	892	1117	1199
With 1 mmol K	773	945	988
With 2 mmol K	739	918	958
With 3 mmol K	717	852	966
With 5 mmol K	702	830	931
With 10 mmol K	700	809	905
With 12 mmol K	700	805	908
With 14 mmol K	700	812	894

T_i_, initial temperature. T_m_, peak temperature. T_f_, final temperature.

**Table 2 molecules-28-06779-t002:** XRD characteristic parameters of the residual coke with different carbon conversions.

Sample	Temperature (°C)	Conversion (%)	d_002_ (Å)	L_c_ (Å)	L_a_ (Å)
Without catalyst	842	15	3.50	15.71	31.80
	1059	26	3.51	16.40	35.32
	1111	37	3.50	16.89	35.49
	1144	50	3.51	17.09	36.32
	1172	70	3.49	23.12	44.18
With 5 mmol K	758	18	3.49	13.72	28.00
	792	30	3.49	13.26	30.43
	820	48	3.49	12.96	31.16
	842	65	3.49	9.82	32.16
	851	71	3.52	9.70	32.76

**Table 3 molecules-28-06779-t003:** Texture parameters of residual coke with different carbon conversions.

Sample	Temperature (°C)	Conversion (%)	Surface Area (m^2^/g)	Total Pore Volume (cm^3^/g)	Average Pore Size (nm)
Without catalyst	718	13	1.81	0.008	27.39
	842	15	2.44	0.008	19.95
	1059	26	101.93	0.030	2.71
	1144	50	99.90	0.121	5.08
With 5 mmol K	714	13	4.01	0.013	13.61
	758	18	6.90	0.017	9.48
	792	30	41.58	0.024	3.63
	820	48	72.17	0.054	3.63

**Table 4 molecules-28-06779-t004:** The characteristic temperature of the combustion of residual coke with different carbon conversions.

Conversion (%)	Without Catalyst			Conversion (%)	With 5 mmol K		
	T_i_ (°C)	T_m_ (°C)	T_f_ (°C)		T_i_ (°C)	T_m_ (°C)	T_f_ (°C)
0	378	523	560	0	378	523	560
15	522	626	680	18	343	506	525
50	523	633	686	48	343	501	535
70	515	626	684	65	340	481	600

T_i_, initial temperature. T_m_, maximum temperature. T_f_, final temperature.

**Table 5 molecules-28-06779-t005:** Proximate and ultimate analyses of the petroleum coke.

Sample	Proximate Analysis (wt.%, ad)				Ultimate Analysis (wt.%, daf)				
	M	A	V	FC	C	H	O *	N	S_t_
ZH	0.28	0.12	9.98	89.62	92.15	3.93	0.79	1.38	1.75

M, moisture. A, ash. V, volatile. FC, fixed carbon. * by difference.

**Table 6 molecules-28-06779-t006:** Concentration of the catalyst in the petroleum coke.

Sample Name	Mass Ratio of K_2_CO_3_ on per Gram of ZH/(g/g)	Molar Ratio of K^+^ on per Gram of ZH/(mmol/g)	Mass Fraction of K_2_CO_3_ on per Gram of Blends (ZH + K_2_CO_3_)/(g/g)
Without catalyst	0	0	0
With 0.1 mmol K	0.0069	0.1	0.69%
With 0.5 mmol K	0.0345	0.5	3.33%
With 1 mmol K	0.069	1	6.45%
With 2 mmol K	0.138	2	12.13%
With 3 mmol K	0.207	3	17.15%
With 5 mmol K	0.345	5	25.65%
With 10 mmol K	0.69	10	40.83%
With 12 mmol K	0.828	12	45.30%
With 14 mmol K	0.966	14	49.14%

## Data Availability

The data presented in this study are available on request from the corresponding author.

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
