# Peer review of "Catalytic Gasification of Petroleum Coke with Different Ratios of K2CO3 and Evolution of the Residual Coke Structure"

_molecules, 2023, doi:10.3390/molecules28196779_

Round 1

Reviewer 1 Report

The manuscript explored the effect of the amount of K2CO3 in the catalytic gasification of petroleum coke and characterized the structure of residual carbon. The work is well-planned and well-executed. However, the following points need to be considered before accepting the manuscript for publication.

 1.     Various ratios of K2CO3: Coke were used for the study. In the 14mmolK, K2CO3 represented 49% of the mass in Coke. So, the catalyst word after K2CO3 throughout the manuscript should be deleted. Further, I suggest renaming the manuscript title as “Catalytic gasification of petroleum coke with different ratios of K2CO3 and the evolution of the residual coke structure”

2.     A table should be provided with the mass of K2CO3 and Coke to prepare the reaction mixture in experimental section 3.1 for a better understanding of the samples.

3.     In figure 1(a) TGA analysis, the residual mass in sample 14mmolK, is around 20%. However, the mass of K2CO3 is around 49%. How does the residual weight come below 49%? The authors should carefully check the analysis and correct it accordingly for all the analysis.

4.     The caption should properly explain the figure. Consider rearranging the caption with more details about (a) and (b) in Figure 1. The same should be followed throughout the manuscript.

5.     What is the meaning of % (15%, 26%, 37%, 50% and 70%) in figure 2. It should be clearly indicated. The same should be corrected in the figure 4, 6 and 8

6.     The caption in Figure 6 is totally misrepresented. The N2-sorption isotherm is a pore-size distribution curve. The pore size distribution curve in the inset of Figure 6 should be separately given for better visibility.

7.      The H3 hysteresis loop of the N2 sorption curve (line no 273) cannot be a generalized statement. The H3 hysteresis loop is only shown in Figure 6(a), 50% without catalyst. What can be the reason for uniform pore distribution in Figure 6a, i.e. without catalyst compared to with catalyst in Figure 6b? This should be discussed in the respective section.

8.     Details of the abbreviation should be given in the first appearance. Details of M, A, V, FC etc. of Table 5 should be given.  

The manuscript should be thoroughly revised and the English language should be checked properly before submission of the revision.

Reviewer 2 Report

Manuscript: molecules-2589399-peer-review-v1

Title: Catalytic gasification of petroleum coke with different ratios of K2CO3 catalyst and the evolution of the coke structure

Amid the vey lengthy manuscript, the paper does not have a very clear thesis. The paper is majorly about describing the equipment, procedures and experimental data collected. There is no clear framing of the work done in relation to an existing problem to be solved. Having the facility to conduct experiments and collect data is one important aspect of research, but this may not be sufficient to grounds for a journal publication considering the facilities used are not very rare. The paper focused only on examining the catalysis “…with different ratios of K2CO3 catalyst” as the title says. Varying the catalyst loading is part of a design of experiment, but this is not enough to have a comprehensive research work fit for a journal paper publication in this research area. There are published papers out there in the same research area and used almost the same equipment but pushed the research by framing their study questions on key aspects of the challenges such as: reaction kinetics formulation from experimental data, reaction mechanisms formulation based on experimental data, influence of catalyst type (MgO, Fe2O3, CaO, Fe3O4, etc.). An example paper: “Catalytic gasification reactivity and mechanism of petroleum coke at high temperature”, 2021, DOI: https://doi.org/10.1016/j.fuel.2021.120469.

The Abstract of the paper is very confusing. First, it does not correctly state the “ratio” of K2CO3. There is a mention of “e K+ ratio higher than 5 mmol” in line 16, but this does not sound correct in the basic definition of ratio. Ratio of what to what: ratio  = (variable 1)/(variable 2)?

The paper also has many sentences that are too long while cramming many information such as lines 14-17 in the Abstract: “The initial peak and final gasification temperature of the petroleum coke decreased greatly with the increase of the K2CO3 catalyst, and the catalyst saturated when the K+ ratio higher than 5 mmol after which the gasification rate varied quite slightly, but no inhibition effect observed.” Please construct sentences with simple and clear meaning and please do this in the whole paper.

Language and grammar: please revise the paper with correct use of grammar. Here is another example in lines 68-69: “The catalytic effect of K2CO3 during the catalytic gasification of petroleum coke is the resultant of many factors…”. Do you mean “result” in place of “resultant”? Another example in line 91: “Considering the above, …”. This is not a complete phrase. Another example of grammar error is line 70: “Though high temperature is benefit for the endothermic gasification…”. Another one is line 74: “Moreover, potassium would vaporized and released…”. These examples of grammar errors significantly reduce the readability of the paper. This kind of error is all throughout the paper as evident in how frequent they appear in the paper (notice the line numbers mentioned above), so please re-check the language and revise for better readability.

Overall, the paper is not in a form that is ready for a journal publication. The paper has some interesting data, but there is no clear framing of the work in the context of the pertinent questions to be solved in the research area.

Major language editing needed. Please see review comments on language and grammar.

Reviewer 3 Report

The authors studied the gasification of petroleum coke in the presence of homogenous potassium carbonate catalyst by a thermogravimetric analyzer (TGA) using the non-isothermal method. Homogeneous catalyst enhanced gasification of petroleum coke is not a new concept however using a TGA analyzer presents new insights especially in the presence of homogeneous catalyst.  There are some issues that needs to be addressed in the manuscript.

- include quantitative information in the abstract 

- There are several grammatical errors in the manuscript 

- In the introduction a table of previous studies related to the gasification of petroleum coke and their limitations should be provided 

- What was potassium carbonate choose? It has corrosive effects and it is soluble in water also there are some also promising mixed oxide catalyst.

- In the results and discussion section

- Description of figure 1 is not clear. The authors need connect the results with DTG to confirm that the mass loss was caused by residual volatiles 

- Can mass balance of solid, liquid and gaseous products be supplied at various temperatures?

- In the methodology the authors should explain when the petroleum coke was collect 

- How was catalyst concentration prepared ?

- Schematics of the entire reactor setup as well as details of analytical characterization should be provided. 

  •  

Needs improvement 

Round 2

Reviewer 1 Report

The authors clearly addressed all the queries and the manuscript is substantially improved.  The manuscript is recommended for publication in its present form. 

Author Response

Thank you for considering our revised manuscript for publication. We are grateful to you and the reviewers for the valuable suggestions provided.

Reviewer 2 Report

Paper Title: Catalytic gasification of petroleum coke with different ratios of K2CO3 and evolution of the residual coke structure

The paper is not a research paper that way it is currently written. Here are the reasons why:

(1)    There is no clear identification of the problem it is trying to solve. Is the work aiming to improve the product gas (e.g., quality of syngas) or to improve the residual coke? Either product stream, there is a need for a clear statement of such goal. Both product streams must be assessed, so if one product stream is favored, say the residual coke, then justify why. Describing your methodology as it is now in the current version of the paper does not accomplish this main component of a research paper. So, what is the problem this work specifically aims to solve?

(2)    There are no clear metrics to measure the performance of the catalysis process. This is significantly dependent on the Issue #1 explained above. If the product gas stream is the focus, then show metrics on how the catalysis is improving the quality of the product gas. If the product residual coke is the focus, then show metrics on how the catalysis is improving the quality of the residual coke.

(3)    The paper does not do anything novel compared to the existing literature on the topic. Like I said in the round 1 of review, there are several papers already published that performed more comprehensive work on catalytic gasification of coke (e.g., “Catalytic gasification reactivity and mechanism of petroleum coke at high temperature”, 2021, DOI: https://doi.org/10.1016/j.fuel.2021.120469). What does this paper bring to the literature of the topic within the framework of solving existing challenges of the research area?

Moderate editing of English language required.

Reviewer 3 Report

All comments have been addressed.

Author Response

(The authors gave the same response as above.)
